# Ultrasonic Measurement of Axial Preload in High-Frequency Nickel-Based Superalloy Smart Bolt

**DOI:** 10.3390/s23010220

**Published:** 2022-12-25

**Authors:** Shuang Liu, Zhongrui Sun, Guanpin Ren, Cheng Liao, Xulin He, Kun Luo, Ru Li, Wei Jiang, Huan Zhan

**Affiliations:** 1Department of Applied Physics, College of Mathematics and Physics, Chengdu University of Technology, Chengdu 610059, China; 2Chengdu Development Center of Science and Technology of CAEP, Chengdu 610299, China

**Keywords:** axial preload, ultrasonic measurement, nickel-based superalloy, bolt preload, high-frequency piezoelectric thin-film sensor, non-destructive testing

## Abstract

A high-frequency, piezoelectric thin-film sensor was successfully deposited on a nickel-based superalloy bolt by radio frequency magnetron sputtering to develop a smart, nickel-based superalloy bolt. Ultrasonic response characterization, high accuracy, and repeatability of ultrasonic measurement of axial preload in nickel-based superalloy smart bolts are reported here and were fully demonstrated. The axial preload in the nickel-based superalloy smart bolt was directly measured by the bi-wave method (TOF ratio between transverse and longitudinal-mode waves) without using the traditional integration of a longitudinal and shear transducer. A model concerning the bolt before and after tensioning was established to demonstrate the propagation and displacement distribution of the ultrasonic waves inside a nickel-based superalloy smart bolt. The measured A-scan signal presented significantly favorable features including a mixture of transverse and longitudinal mode waves, a pure and broad frequency spectrum which peaked at 17.14 MHz, and high measurement accuracy below 3% for tension of 4 kN–20 kN. For the temporal ultrasonic signal, the measurement envelopes were narrower than for the counterpart of the simulation, justifying the ‘filtration’ advantage of the high-frequency sensor. Both the TOF change of the single longitudinal-mode wave and the TOF ratio between transverse- and longitudinal-mode waves increased linearly with preload force in the range of 0 kN to 20 kN. Compared with the commercial piezoelectric probe, the proposed probe, based on the combination of a high-frequency, piezoelectric thin-film sensor and a magnetically mounted transducer connector, exhibited high tolerance to temperatures as high as 320 °C and high repeatability free from some interference factors such as bolt detection position change and couplant layer thickness. The results indicate that this system is a promising axial preload measurement system for high-temperature fasteners and connectors, and the proposed sensor is a practical, high-frequency ultrasonic sensor for non-destructive testing.

## 1. Introduction

Nickel-based superalloys are attractive materials for the components used in the hot zones of jet turbine engines and other industrial applications where exceptional resistance to high-temperature working conditions is required [1,2,3]. As bolts are critical, high-temperature components, nickel-based superalloy bolts are a credible choice for top-notch, rugged applications in aircraft, aerospace, oil and gas, and various industries [1]. The accurate measurement and monitoring of the axial preload of bolts are critical for maintaining important infrastructure performance and preventing premature failure accidents [2,3,4,5,6]. On this basis, a lot of attention has recently been paid to research in the field of structural health monitoring and the measuring of bolts and bolted connections under the influence of loss of load capacity and high-temperature working conditions [2,3,4,5,6,7,8,9,10]. There are many ways to directly or indirectly measure bolt preload [2,4,5,6,7,8,9,10,11,12,13,14,15]. Direct measurement methods are mainly strain gauges [14] and torque control [15]. These methods are employed less due to their low accuracy and significant error, and they do not allow for the online monitoring of bolted structures. This has pushed detection and investigation methods toward indirect methods [7,8,9,10,11,12,13,14,15,16,17]. Many indirect methods such as impedance-based methods [16], piezo active sensing methods [17], acoustoelastic-based methods [7,8,9,10], etc., have been intensely investigated in recent years and have been proved to evaluate bolt preload force. Some detection methods can basically realize unmanned online monitoring. However, most indirect methods have different intrinsic shortcomings. For instance, impedance-based methods [16] suffer from relatively high cost, large size of setup, and sensitivity to thermal and load fluctuations, although they are highly sensitive and suitable for in situ monitoring. 

Among these indirect methods, acoustoelastic-based methods, i.e., ultrasonic preload measurement, combine the attributes of high precision, excellent real-time performance, and strong sensing penetration and, therefore, have been the most widely used for the detection of axial preload in bolts [2,4,7,8,9,10,11,12,13]. Ultrasonic preload measurement methods can be roughly classified into the mono-wave method and bi-wave method according to the nature of the measurement [2,4]. The mono-wave method, proposed earlier, refers to the use of a single transverse wave or longitudinal wave to measure the axial stress in a bolt [2,4,7,11]. This method is generally used to control and verify the preload applied during the assembly of the bolt [2,4]. For comparison, the bi-wave method relies on a combination of transverse- and longitudinal-mode waves to measure the axial preload in bolts [18]. The method provides easier detection of the axial preload in an already tightened bolt [2,4,18,19]. Currently, to achieve the bi-wave method for bolt measurement, a transverse- or longitudinal-mode wave probe is adopted to collect ultrasonic wave signals [18,19]. Another way is to integrate the transverse- and longitudinal-mode wave probe together for the bi-wave method. These ways not only have intricate measurement and large-size probe fabrication processes, but they also easily cause additional measurement errors in terms of axial preload during the measurement process.

Prior to measurement, to avoid error caused by the repeated coupling of transducers, the end face of the subject bolts is processed by some methods such as turning and polishing [7]. However, it not only damages the bolt body but also consumes a great deal of time and labor. These limitations can be overcome by moving to an electromagnetic acoustic transducer (EMAT) [2] and a permanently mounted transducer system (PMTS) [20]. They have been gradually replacing the piezoelectric transducer in recent years. However, the two transducers have not been widely used in engineering applications relating to bolt preload or stress. For example, the use of an EMAT for bolt preload measurement is limited by low conversion efficiency and difficulty in exciting the longitudinal wave on the end faces of ferromagnetic bolts [2,4,20]. The special handling process of the bolt and lead content frustrates the wide use of PMTS [2]. Similar to PMTS, an ultrasonic sensor deposited or bonded on the bolt surface has been demonstrated recently [7]. The common mechanism of the ultrasonic sensor action is described in Figure 1. Based on the inverse piezoelectric effect, the excitation electric pulse signal is easily converted into an ultrasonic wave signal which propagates along the axial direction of the bolt and generates the echo at the bottom; subsequently, the echo propagates along the axial reverse direction and reaches the ultrasonic sensor, and the echo is converted into an electrical pulse signal with preload- or stress-related information through the piezoelectric effect. Based on a smart piezoelectric sensor bolt with a frequency constant of 1.89 MHz, Q. Sun et al. [7] recently reported precise bolt preload measurement by only the mono-wave method. However, the piezoceramic transducer was fixed in the center of the bolt head by low-temperature-tolerance (150 °C) epoxies. Currently, most ultrasonic-related sensors and probes are low frequency and have low temperature tolerance due to the couplant requirement and single-mode wave transducers below 10 MHz [2,4,7,8,9,10,11,12,13,14,15,16,17,18,19,20]. To the best of our knowledge, little literature has so far reported on the axial preload measurement of non-ferromagnetism or weak-ferromagnetism smart bolt specimens such as nickel-based superalloy and titanium alloy bolts. There have been almost no reports on high-frequency, piezoelectric ultrasonic sensors for the axial preload of bolts before.

In this work, smart, nickel-based superalloy bolts with a high-frequency, piezoelectric thin-film sensor were successfully fabricated by radio frequency magnetron sputtering. High accuracy and repeatability of ultrasonic measurement of axial preload for the smart, nickel-based superalloy bolts were fully demonstrated. Thanks to the proposed probe, i.e., the combination of a high-frequency, piezoelectric thin-film sensor and a magnetically mounted transducer connector, the collected ultrasonic wave signals possessed pure and broad frequency centered at 17.14 MHz and included both transverse- and longitudinal-mode waves. These facts enabled us to measure the axial preload in the nickel-based superalloy by either the mono-wave method or the bi-wave method. Compared with the simulated ultrasonic response, the measured ultrasonic response showed narrower envelopes due to the ‘filtration’ character of the proposed sensor. The pure ultrasonic response allowed a small preload of 800 N to be successfully measured with the introduction of a spline interpolation algorithm. The proposed high-frequency probe showed obvious advantages over the commercial piezoelectric probe in terms of frequency spectrum, high temperature tolerance, small preload measurement, repeatability, measured error caused by couplant layer thickness, and measurement position change.

## 2. Research Methods

### Measurement and Experiment Process

Ultrasonic pulse-echo technology based on acoustoelasticity theory was used here to measure the axial stress in a smart bolt [2]. For the mono-wave method, axial preload in the nickel-based superalloy bolt was calculated by the TOF change measurement of the pulse echo before and after tightening [2,4]. The relationship between the TOF change and axial preload force can be expressed as follows [2]:(1)$$F=\frac{E\cdot S\cdot \Delta L}{L}$$
where F is the axial preload force, ΔL=Δx·V is the change of bolt length, Δx is the change in TOF from before to after tightening, and V is the propagation speed of the ultrasonic longitudinal wave in the material; E is the elastic modulus of bolt material, S is the effective cross-sectional area, and L is the clamping length of the bolt.

An integrated excitation and collection ultrasonic system based on a high-frequency piezoelectric sensor was constructed and is depicted in Figure 1. This proposed system consists of an ultrasonic signal excitation and acquisition integrated module, a computer with measurement software, and an ultrasonic probe. The independently developed excitation and acquisition integrated unit could not only launch the exciting voltage in the range of 0~400 V but could also collect ultrasonic wave signals with a frequency from 0.2 MHz to 25 MHz. The corresponding sampling frequency was 100 MHz, and the maximum gain value was as high as 89 dB. The analog signal filter in the frequency domain was from 2 MHz to 25 MHz in this work. The influence of temperature on the preload measure was not considered here.

Different from the commercial piezoelectric probe shown in Figure 2a, the ultrasonic probe consists of two disconnected parts: a high-frequency, piezoelectric thin-film sensor and a magnetically mounted transducer connector, as shown in Figure 2b. To address the disadvantages of the ultrasonic probe, the high-frequency, piezoelectric thin-film sensor was deposited on the slightly polished surface of the nickel-based superalloy head, as shown in Figure 2c. The smart bolts that were the specimens in the experiment were M8 nickel-based superalloy bolts with four thin-film, multi-function layers: an electrode layer, isolated layer, high-frequency, piezoelectric thin-film layer, and transition layer. The composition of the four layers was GH4 169, ZnO, Cr_2_O_3_, and Sn, respectively. The ZnO ceramic target was a high-frequency piezoelectric material with a high purity of 99.99% in this work. The deposition processes were performed inside radio frequency magnetron sputtering [21], as shown in Figure 2d. The fabrication procedure was as follows: Through a nanosecond pulsed laser polishing technique [22], the surface roughness of the nickel-based superalloy bolt head was controlled below 0.4 μm, and the parallelism between the top and the bottom of the bolt was controlled at 0.01 mm. Then, the nickel-based superalloy sample was consecutively cleaned by an acetone, ethanol, and deionized water solution through an ultrasound cleaner and dried by high-purity nitrogen; to remove the surface pollutants of the target, it was required to pre-sputter the high-frequency piezoelectric target for 20 min. After that, the sputtering experiment was started; by radio frequency magnetron sputtering method, four thin-film, multi-function layers were gradually deposited on the polished surface of the nickel-based superalloy bolt head. The process parameters for preparing the high-frequency thin films on the surface of the nickel-based superalloy bolt were as follows: the base vacuum pressure was 8.4 × 10^−4^ Pa, the sputtering pressure was 1.0 Pa, the sputtering power was 250 W, the ratio of argon to oxygen flow rate was 10:2, the distance between the target and the substrate was 7 cm, the rotation rate of the sample stage was 5 r/min, annealing for 1 h in oxygen atmosphere, the annealing temperature was 400 °C, and the sputtering time was 40 min. To avoid the introduction of additional stress, the thin-film deposition was performed at room temperature. The thickness of the whole thin film was ~14 μm. The thickness of the electrode layer, isolated layer, high-frequency, piezoelectric thin-film layer, and transition layer was 5.2 μm, 6.0 μm, 2.1 μm, and 1.1 μm, respectively. To the protect the high-frequency thin-film layer from corrosion and pollution, an isolated metallic oxide layer was introduced between the electrode layer and the high-frequency thin-film layer. The material property of the thin-film sensor is shown and discussed in the following sections. 

## 3. Result and Discussion

### 3.1. Theoretical Model and Simulation Results

To simulate the propagation process of ultrasound inside the nickel-based superalloy bolt, numerical calculation was carried out thoroughly with the well-known finite element method that is reported elsewhere [23,24]. The expression of the analytical function of the electrically excited bolt is [25]:(2)$$an1\left(t\right)=V_{0}*sin\left(A*t\right)*gp1\left(t\right)$$
where *V*_0_ is the excitation voltage, *A* is the amplitude, *gp*1(*t*) is the Gaussian pulse, and the function expression of the Gaussian pulse is [26]:(3)$$gp1\left(t\right)=A*e^{-\frac{2\left(t-2T_{0}\right)}{T_{0}}}$$
where *A* is the amplitude, and *T*_0_ is the period.

In the piezoelectric effect, the deformation of the piezoelectric film is caused by the voltage applied by the electric excitation. The mechanism of the piezoelectric effect can be summarized as the following governing equation [25]:(4)$$\rho \frac{\partial^{2}u}{\partial t^{2}}=\nabla \cdot S+Fv$$
where *ρ* is the material density, and *u* is the displacement field. *S* is the stress tensor, and Fv is the deformation gradient.

Table 1 gives the input parameters of the acoustoelasticity model for the smart bolt in the work. Figure 3 shows the displacement distribution of the ultrasonic wave along the whole bolt in the axial direction for cases of 0 N and 20 kN. To simplify the simulation result, we only considered elongation of the bolt length induced by the axial preload, and, therefore, the length of the bolt mode lengthened by 0.233 mm according to the experiment assessment, as shown in Figure 3. An ultrasonic wave was produced on the upper surface of smart bolt through 60 V voltage excitation and directly transmitted toward the bottom of the bolt at time 2 μs. The trailing waves could obviously be seen at time 2–6 μs, and part of the ultrasonic wave was reflected by the boundary of the bolt side. The ultrasonic wave reached the boundary of the bolt bottom at time 6 μs and then completed the reflection behavior on the boundary at time 8 μs. One part of reflection wave energy transmitted in the opposite axial direction. It was observed, from 12 μs to 14 μs, that the ultrasonic wave was gradually detected at the bolt head to generate the first longitudinal wave. Compared with the slack bolt, the bolt to which 20 kN preload had been applied presented a similar ultrasonic propagation process with a corresponding time decay. Figure 4a shows the simulation results of the temporal ultrasonic signal before and after tensioning at 20 kN. With the introduction of 20 kN tension, the whole temporal ultrasonic signal presented a delay phenomenon, and a TOF change for the first longitudinal wave was observed. The absence of a shear wave in the temporal signal is attributed to the pressure acoustic module of the used COMOSOL Multiphysics software, which only contains longitudinal wave mode. The displacement field distribution of the upper surface for the slack bolt and the preload bolt with 20 kN was calculated and presented in Figure 4b. Two peak components of the displacement field were found to correspond to the first and second longitudinal wave. The axial preload of 20 kN had no remarkable influence on the displacement field intensity of the longitudinal wave, which led to a time delay.

### 3.2. Measured Ultrasonic Wave Properties

Spline function interpolation was adopted to increase data points and improve the resolution of the propagation time difference [26]. Figure 5a shows the measured A-scan waveform collected from the nickel-based superalloy bolt head. The first longitudinal wave was the envelope, which peaked at 14.10 μs, corresponding to the time of one round trip of the longitudinal wave propagating along the whole bolt in the axial direction. Likewise, the second longitudinal wave corresponded to the time of two round trips of longitudinal wave propagation, and the envelope was centered at 27.95 μs with a reduction of voltage amplitude. The intense envelope between the first and second longitudinal waves was the first shear wave, peaking at 26.85 μs. The unique feature of the coexisting longitudinal and shear wave made us directly use the bi-wave method for the preload measurement. Until now, rare ultrasonic transducers have been reported to generate longitudinal and transverse waves simultaneously. In addition to this, it was noted that the voltage amplitude of the first shear wave was more intense than the counterpart of the longitudinal wave. The reason may be attributed to the piezoelectric thin-film sensor possessing a more intense poling phenomenon at the plane direction compared with the perpendicular direction when the thin-film layer was excited by 60 V voltage. Limited by the experimental conditions, however, the corresponding microscopic mechanism could not be effectively obtained in this work. Figure 5b compares the TOF of the pulse echo before and after applying tension at 20 kN. The use of tension led to an obvious TOF change and a reduction of amplitude intensity. Nevertheless, the profile of the curve presented no change. In this case, the cross-correlation algorithm [9] was used to obtain the TOF change before and after tension. Compared with the simulation results (see the Figure 4), the experimental results (see Figure 5a,b) presented more narrow envelopes. As the sensor intrinsic material property could not be considered during the simulation result, the reflected waves and the scattering waves from other surfaces were mixed into the echo signal. However, the measured narrow envelopes could be filtered by the high-frequency sensor. The intrinsic reason was unclear. For the outline and time position, the experimental results basically coincided with theoretical simulation result. To check the high temperature tolerance, the smart nickel-based superalloy bolt was inserted into a heating furnace with a temperature of 320 °C and kept at a continuous heat for 1 h. As shown in Figure 5c, the whole temporal ultrasonic signal at 320 °C also presented time decay comparable to its counterpart at 22 °C, while the amplitude intensity presented no significant reduction. It suggests that the smart bolt can tolerate a high temperature of 320 °C without failure. The further high-temperature test was limited by the tolerance temperature (≤250 °C) of the available BNC connector line. For comparison, ultrasonic measurement of the bolt was carried out by replacing the magnetically mounted transducer connector with a commercial piezoelectric probe centered at 10 MHz. As indicated in Figure 5d, the temporal ultrasonic wave signal presented some obvious background noise and no shear mode wave between the first and second longitudinal waves. The envelope of the first peaked longitudinal wave of the commercial probe was obviously wider than the counterpart of the smart bolt. It validated the ‘filtration’ advantage of the high-frequency piezoelectric sensor with weak interference from the reflected waves and the scattering waves from other surfaces.

Frequency spectra of the first longitudinal waves collected from the commercial probe and high-frequency piezoelectric sensor were obtained by the Fastest Fourier Transform in the West (FFTW) algorithm. The results are shown in Figure 6. For the commercial probe, in addition to the frequency centered at 8.32 MHz, several other harmonic waves could be easily observed, as shown in Figure 6a. These harmonic waves originated from nonlinear ultrasonic effects, including sum-frequency and difference-frequency effects [27], and the piezoelectric material intrinsic property [28]. An increase in tension led to a more intense inhomogeneous stretch along the axial direction of the bolt. Owing to nonlinear ultrasonic effects, energy transfer between low- and high-frequency signals occurred continually with the increase in the tension. In this case, the similarity of the ultrasonic wave signals before and after straining was reduced, so the calculated results by the cross-correlation algorithm were difficult to keep at high accuracy. By comparison, the first longitudinal-mode wave collected by the piezoelectric sensor corresponded to a pure and broad spectrum centered at 17.14 MHz, as shown in Figure 6b. Neither a low-frequency peak nor a high-frequency peak was observed. For the smart bolt reported in [7], the center frequency of the piezoelectric sensor was 1.89 MHz, and no frequency spectrum was presented. As reported in [11], the frequency spectra presented obvious higher-harmonic-related components. With the increase in the tension, the spectrum center showed no obvious shift, although the amplitude intensity presented a slight reduction. It justifies the advantages of the piezoelectric sensor with a high nonlinear suppression feature and almost no energy transfers of different frequency signals. This ensures that mono-wave methods based on the cross-correlation algorithm are capable of achieving a precise measure of the bolt preload.

### 3.3. Axial Preload Measurement

During the measurement experiment, a specially designed clamp was used to connect the nickel-based superalloy smart bolt specimens together with a magnetic connector to form a whole body. After that, axial tension was applied from 0 to 20 kN with an interval of 4 kN by the calibrated electrical universal material testing machine. In the experiment, the repeatability error for 4 kN, 8 kN, 12 kN, 16 kN, and 20 kN was calculated to be 0.42%, 0.37%, 0.36%, 0.27%, and 0.22%, respectively. Synchronously, the ultrasonic measurement system excited and collected ultrasonic wave signals by computer software. For the five bolt samples, axial preload was measured experimentally and linearly fitted as a function of the TOF change, as shown in Figure 7a. For each sample, actual measured tension values and fitted tension values had a very high fitting precision, i.e., a good linear relationship between the TOF change of the first longitudinal wave and tension. Neither a nonlinear curve nor some deviation points were observed. This validates the effectiveness of the TOF change calculated by the cross-correlation algorithm. To examine the feasibility of the bi-wave method, the TOF ratio of the first longitudinal wave to the first shear wave as a function of the tension was measured and is shown in Figure 7b. A good linear relationship between the tension and TOF ratio of the first longitudinal wave to the first shear wave was also observed. It means the in-service bolt preload could be directly measured by the calibration curve function shown in Figure 7b. For the in-service superalloy smart bolt, the proposed measurement system with only a magnetic connector directly measured its axial preload only one time due to the simultaneous generation of the longitudinal and shear waves. Compared with the previously reported bi-wave methods [18,19], the bi-wave method based on the high-frequency sensor is more effective and accurate for measuring axial preload of in-service bolts. The coefficient of determination R^2^ for the bi-wave method was 0.99878, which is in agreement with that of the mono-wave wave method. Therefore, axial preload of the installing, already tightened, or in-service superalloy smart bolts is capable of being effectively and accurately measured by the proposed ultrasonic measurement system based on a high-frequency sensor. 

In the experiment, a heat module installed on an electrical universal material testing machine was inconvenient and prohibited for safety. As the change in temperature from room temperature to 250 °C was large, many parameter calibrations were complicated and limited by the presented experiment condition. Considering these, the relationship between the temperature and TOF in an unstressed bolt was measured as shown in Figure 7c. TOF in unstressed sample 5 was firstly recorded at 20 °C. Sample 5 was inserted into a heap of sandy soil sustainably heated by the furnace and kept for half an hour at each measurement temperate. TOF of the heated sample 5 was quickly measured and recorded. This process was repeated until the temperature reached 250 °C. The temperature increased in 10 °C steps. It showed a nearly linear relationship between the temperature and TOF of sample 5. The coefficient of determination R^2^ was 0.99761. The results could be used as compensation for the actual high-temperature tension measurement [2] in the following work. To examine the ability to measure the small preload below 1 kN, a small axial preload of 800 N was applied to sample 5 by an electrical universal material testing machine. The measured temporal ultrasonic signal curves are shown in Figure 7d. As shown in the inset of Figure 7d, no TOF change was observed for the original temporal signal for the slack and small preload due to the limited sampling frequency of 100 MHz. The spline interpolation algorithm was used to increase time resolution. After that, the preload value measured by the proposed system fluctuated in the range of 0.75~0.84 kN, corresponding to an error of less than ±7%. The ultrasonic signal amplitude of the slack bolt was slightly lower than that of the bolt with 800 N applied, as described in Figure 7d. Both the value fluctuation and the lower amplitude of the slack smart bolt may be attributed to the small inner noise interference of the acquisition unit and different contact pressure between the magnetically mounted transducer connector and the smart bolt.

### 3.4. High Accuracy and Repeatability Property

To identify the advantage of the high-frequency sensor for preload measurement, five nickel-based superalloy bolts were chosen as experimental specimens and loaded with tension from 4 kN to 20 kN by the calibrated testing machine. The measurement error was obtained from a function (measure error σ = 100% ∗ (F_measure_ − F_apply_)/F_apply_, where F_measure_ is the tension value measured by the software, and F_apply_ is the tension value applied by the testing machine). The calculation results of measurement error are shown in Figure 8a. It was noted that the absolute error of this system was below ±0.3 kN, and the relative error was less than ±3%. The error results were in agreement with the counterpart reported in [7]. As a nickel-based superalloy bolt is a kind of extremely weak magnetic material, insufficient contact between the magnetic connector and the smart bolt was easily induced during the stretching test process. The low ultrasonic preload measured error for the weak-magnetism bolt justifies the advantage of the high measurement accuracy of the proposed measurement system. Compared with a carbon-steel-based bolt [7], the superalloy smart bolt with weak magnetism easily led to deficient contact and was excited by external voltage. As a result, the excited position of the sensor easily changed the required uniform piezoelectric property around the center region of the sensor. Additionally, the contact position change easily caused the angle deviation between the excitation and collection of ultrasonic wave. For a carbon-steel-material smart bolt, the absolute error of the system was less than ±0.2 kN (corresponding to <±1%) in the experiment because the magnetic connection contributed to the fixed contact position point and both the excitation and collection of the ultrasonic wave. These justify the high accuracy of the ultrasonic measurement system based on a high-frequency piezoelectric sensor. 

To examine the high repeatability of the proposed sensor, the magnetic connector was mounted at ten different positions on the nickel-based superalloy bolt head surface. The corresponding ten ultrasonic measurements were carried out. The measurement results are shown in Figure 8b. All of the first longitudinal mode signals almost overlapped together. This means the proposed ultrasonic system is insensitive to the contact position. This is due to the vibration–electrical energy conversion occurring between the four thin-film layers, as shown in Figure 2c. For comparison, the commercial piezoelectric probe could be used to measure ultrasonic wave signal at two different contact positions and coupling layer thicknesses. These measurement results are presented in Figure 8c,d. Obvious deviation phenomena between the ultrasonic wave signals could be easily seen in the cases of different contact positions and coupling layer thicknesses. This drawback makes it difficult for the commercial piezoelectric probe to measure the axial preload in actual engineering applications.

## 4. Conclusions

In summary, newly developed nickel-based superalloy smart bolts with a high-frequency (center frequency: 17.14 MHz), piezoelectric thin-film sensor were fabricated by radio frequency magnetron sputtering. The ultrasonic temporal and spectrum domain response and axial preload measurement of the nickel-based superalloy bolt were fully demonstrated. The proposed piezoelectric sensor possesses a pure and broad frequency spectrum from 10 MHz to 30 MHz, high-temperature-tolerance ability up to 320 °C, and a mixture of transverse- and longitudinal-mode waves. Compared with the commercial piezoelectric probe, the proposed sensor presents stable ultrasonic properties with high suppression of the nonlinear ultrasonic effect. The measured temporal signal presented narrower signal envelopes than the simulation results, indicating the ‘filtration’ advantage of the proposed sensor. Both the mono-wave method and the bi-wave method were applied to measure the axial tension for small preloads from 800 N to 20 kN. TOF presented a linear increase trend with temperature increasing from room temperature to 250 °C. The ultrasonic system based on the smart bolt showed excellent characteristics of high repeatability, low measurement error of axial preload below ±3% (comparable to the counterpart of the previously reported carbon-steel bolt), small preload measurement ability, and high stability. As a fixed contact position point was difficult to keep long enough for weakly magnetic smart bolts, high-accuracy ultrasonic measurement of the axial preload in this work indirectly demonstrated the uniform piezoelectric distribution in the whole high-frequency sensor. Future research work will be transferred to monitor the axial preload of the high-temperature, in-service bolt and establish a high-precision and -frequency laser ultrasonic system based on the high-frequency piezoelectric sensor.

## Figures and Tables

**Figure 1 sensors-23-00220-f001:**
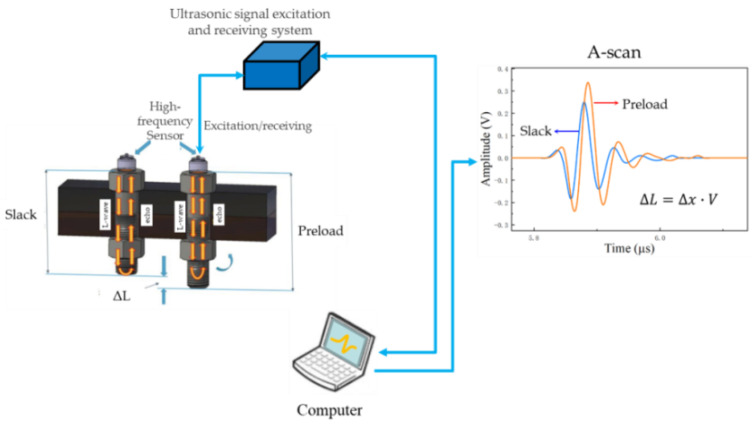
Experimental setup schematic and fundamental principle diagram of measuring bolt axial preload by the mono-wave method. (L-wave: longitudinal wave, V: velocity of longitudinal wave, ΔL, Δx: length and time change of the smart bolt induced by the preload compared with the slack bolt).

**Figure 2 sensors-23-00220-f002:**
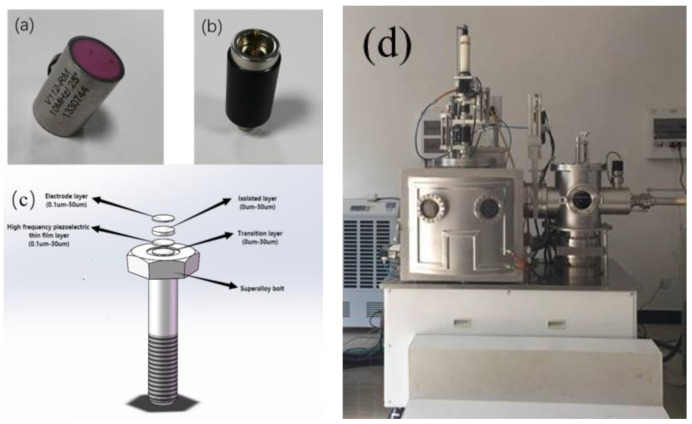
Photographs of (**a**) a commercial piezoelectric probe of 10 MHz and (**b**) a magnetically mounted transducer connector; (**c**) schematic diagram of a high-frequency M8 smart nickel-based superalloy bolt; (**d**) photograph of radio frequency magnetron sputtering.

**Figure 3 sensors-23-00220-f003:**
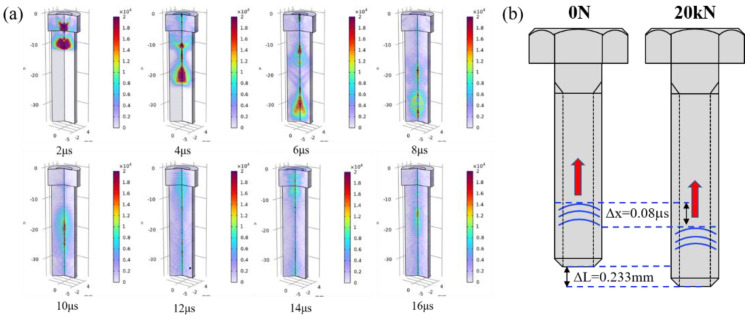
(**a**) Evolution of the ultrasonic wave inside the nickel−based superalloy bolt, (**b**) schematic diagram of ultrasonic wave propagation difference between the slack bolt and the bolt with 20 kN applied.

**Figure 4 sensors-23-00220-f004:**
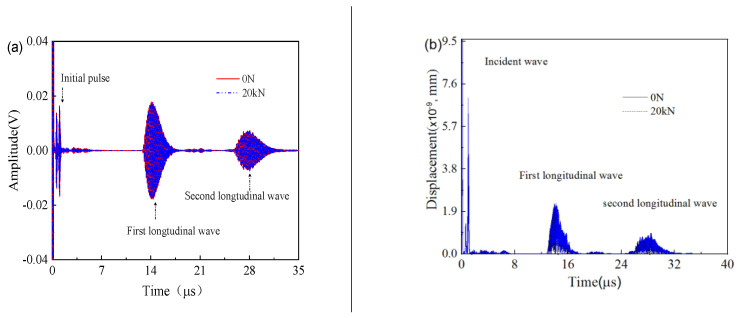
(**a**) The simulated temporal ultrasonic signal inside the nickel−based superalloy bolt, (**b**) displacement field contrast of the upper surface between the bolt with 20 kN axial preload applied and the slack bolt.

**Figure 5 sensors-23-00220-f005:**
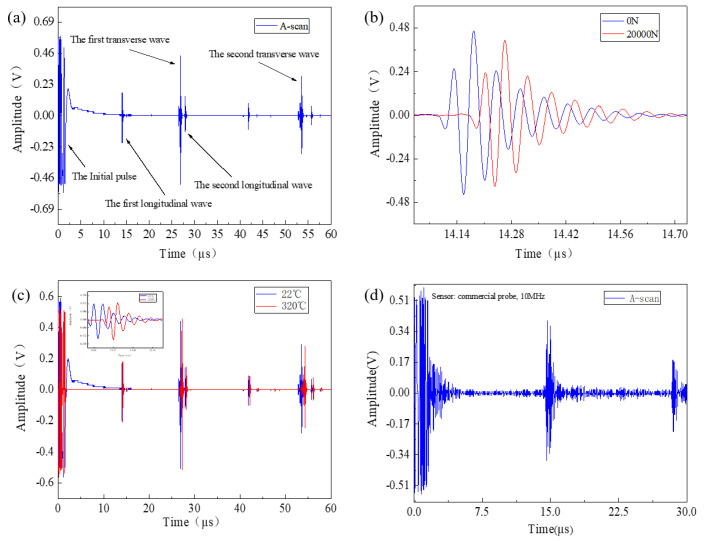
(**a**) The measured A−scan waveform collected from the high−frequency, piezoelectric thin−film sensor, (**b**) comparing TOF change of the pulse echo before and after applying tension at 20 kN, (**c**) A−scan waveforms of the smart bolt at 22 °C and 320 °C (the insert compares the waveform of the first longitudinal wave for the bolt at 22 °C and 320 °C), (**d**) the measured A−scan waveform collected from the commercial piezoelectric probe.

**Figure 6 sensors-23-00220-f006:**
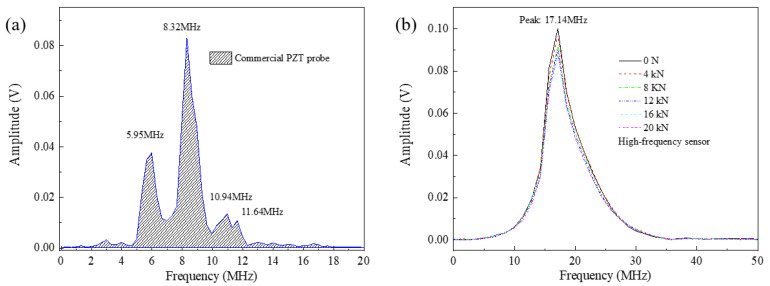
Frequency spectra of the measured ultrasonic wave collected from (**a**) the commercial PZT probe and (**b**) the high−frequency piezoelectric sensor.

**Figure 7 sensors-23-00220-f007:**
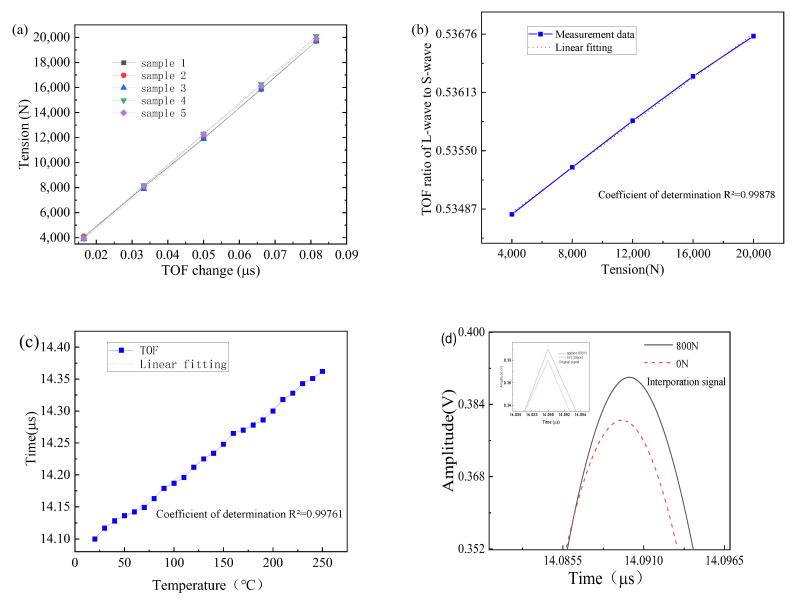
(**a**) Relationship between the TOF difference and the tension; (**b**) TOF ratio of the first longitudinal wave to the first shear wave as a function of the tension; (**c**) relationship between the temperature and TOF in an unstressed bolt; (**d**) TOF change before and after the applied axial preload of 800 N after interpolation. The inset is the original temporal signal of the first longitudinal wave for the cases of 0 N and 800 N.

**Figure 8 sensors-23-00220-f008:**
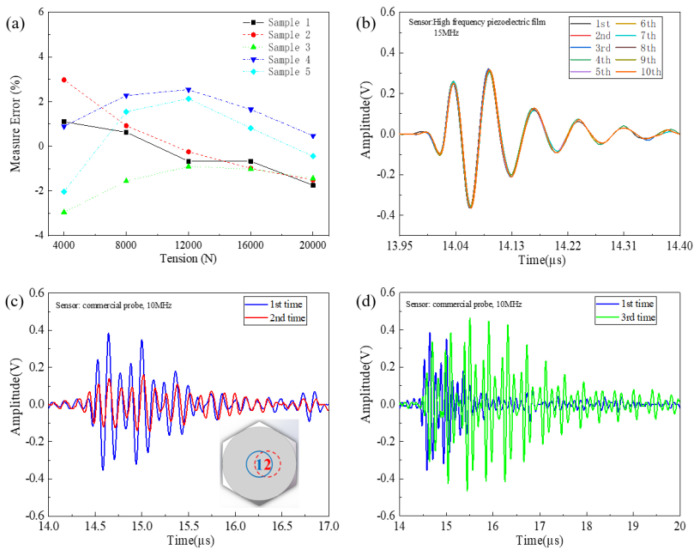
(**a**) The relative error of ultrasonic measurement based on the high-frequency sensor; (**b**) temporal ultrasonic wave signs for ten different contact positions between the magnet connector and smart bolt; (**c**) temporal ultrasonic wave signs for two different contact positions between the commercial piezoelectric probe and smart bolt (inset shows schematic diagram of two different detection positions on the surface of the bolt head.); (**d**) temporal ultrasonic wave signs for two different couplant layer thicknesses.

**Table 1 sensors-23-00220-t001:** Input parameters of acoustoelasticity model for smart bolt.

Parameter	Unit	Value
Longitudinal wave speed	m/s	5788
Frequency, f0	MHz	17
Function, A	2 ∗ π ∗ f0 (MHz)	10.6814
Period, T0	s	5.8824 × 10^−8^
Wavelength, λ0	m	3.4047 × 10^−4^
Exciting voltage, V0	V	60

## Data Availability

Data is contained within the article.

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
