# Peer review of "Ultrasonic Measurement of Axial Preload in High-Frequency Nickel-Based Superalloy Smart Bolt"

_sensors, 2022, doi:10.3390/s23010220_

Round 1

Reviewer 1 Report

Comments to the authors:

1. Avoid group citations in the introduction. [Ex. 1-3, 2-6, 2-10, 7-17, 7-10, 7-13, etc].

2. Research gap and novelty should be mentioned at the end of the introduction.

3. Photograph of the experimental setup has to be added to the article and the experimental procedure is not clear.

4. The influence of temperature on the preload measure wasn’t considered. Why?

5. What is the thickness of the four individual layers? Also, state the influence of varying the layer thickness.

6. All the analytical expressions have to be cited properly.

7. The obtained FEM results have to be discussed with literature support.

8. All the figures need to be cited in the order.

9. There is no information and inference from Figure 6.

10. The authors have to perform an uncertainty analysis and interpret their findings.

11. A detailed discussion section has to be added and discuss the research findings with literature support and bring the essence of the research to the readers.

12. Ref 22, and 25 has to be replaced with recent literature.

13. Few references are not in the format.

14. Kindly avoid the term 'we' and 'our' in the article.

Reviewer 2 Report

In the manuscript entitled “Ultrasonic measurement of axial preload for superalloy bolt based on high-frequency piezoelectric thin-film sensor”, authors have developed a high-frequency piezoelectric thin-film sensor for an accurate and repeatable ultrasonic measurement of axial preload in superalloy bolts.  The 24% similarity rate of the manuscript may look acceptable, however, as per my check most of it is borrowed from the author's previously published article https://doi.org/10.3390/s22228665 and I highly recommend the authors rewrite those parts. Moreover, the novelty of the study should be highlighted for the major readers of the sensor. Moreover, the authors should consider the following comments as well.

1.     Change the title, make it brief and catchy

2.     Add the significance of the study and the novelty of the study to the abstract.

3.     “Axial stress”, “Ultrasonic waves”, and “Nickel-based superalloy” are the three suggested keywords, but they only appeared four times in total in the whole manuscript.

4.     Explain the mechanism of action of the sensor in the introduction.

5.     Section “2.1. Ultrasonic measurement system” is the same as authors' previous publication.

6.     Figure 4, add the experimental temporal ultrasonic signal inside the superalloy bolt for compression with the simulation and discuss the similarities and differences.

7.     Present the mechanical vibration displacement and discuss its results.

8.     Present the ultrasonic longitudinal wave propagating along the axial direction and discuss its results.

9.     Section “3.3 Axial preload measurement”, the original signal before and after applying a small preload should be presented and discussed.

10.  Section “3.3 Axial preload measurement”, the received signals from the ultrasonic sensor during different torque−tightening processes have been skipped.

11.  There is a lack of comparison between the finding of the current study and other similar published literature.

12.  Conclusion is too short and empty of any significant novel achievement.

Reviewer 3 Report

This paper investigates a method for measuring bolt force using high-frequency ultrasound. If detailed and careful revisions can be obtained, it can be further evaluated, otherwise it will be recommended for rejection.

1. In terms of the method, the measurement scheme and model in this paper are not original. I understand that the authors want to highlight the structure of their designed transducer, for example, its structure is more suitable for bolt detection. However, no specific information is given about the design of the sensor. It is suggested that the authors further clarify the innovation point of this paper.

2. The lack of photos of the experimental site and close-ups of the sensor make it difficult to support the innovative nature of the paper.

3. From the method of this paper, the acquisition of TOF is important, which directly affects the uncertainty of detection, and its specific processing method for acoustic signal is not described in this paper.

4. In the experiment, how the authors ensure the accuracy of the bolt force. That is, how is the true value guaranteed?

5. It is recommended that the authors give a detailed plot of all the data in Figure 7a and the processing.

6. Is the method of this paper general for different bolts? For different bolts if pre-experiments are needed to build the model, the method will not have applicability. Correspondingly, the theoretical model of this paper is lacking.

Round 2

Reviewer 1 Report

Dear Authors

I wish you all the best.

Reviewer 2 Report

Good job with the revision.

Reviewer 3 Report

well revision.